# Cbfβ Is a Novel Modulator against Osteoarthritis by Maintaining Articular Cartilage Homeostasis through TGF-β Signaling

**DOI:** 10.3390/cells12071064

**Published:** 2023-03-31

**Authors:** Xiangguo Che, Xian Jin, Na Rae Park, Hee-June Kim, Hee-Soo Kyung, Hyun-Ju Kim, Jane B. Lian, Janet L. Stein, Gary S. Stein, Je-Yong Choi

**Affiliations:** 1Korea Mouse Phenotyping Center (KMPC), Department of Biochemistry and Cell Biology, Cell and Matrix Research Institute, School of Medicine, Kyungpook National University, Daegu 41944, Republic of Korea; 2Department of Orthopedic Surgery, School of Medicine, Kyungpook National University, Kyungpook National University Hospital, Daegu 41944, Republic of Korea; 3University of Vermont Cancer Center, Larner College of Medicine, University of Vermont, Burlington, VT 05405, USA; 4Department of Biochemistry, Larner College of Medicine, University of Vermont, Burlington, VT 05405, USA

**Keywords:** articular cartilage, Runx1/Cbfβ complex, osteoarthritis, TGF-β signaling, proteasomal degradation

## Abstract

TGF-β signaling is a vital regulator for maintaining articular cartilage homeostasis. Runx transcription factors, downstream targets of TGF-β signaling, have been studied in the context of osteoarthritis (OA). Although Runx partner core binding factor β (Cbfβ) is known to play a pivotal role in chondrocyte and osteoblast differentiation, the role of Cbfβ in maintaining articular cartilage integrity remains obscure. This study investigated Cbfβ as a novel anabolic modulator of TGF-β signaling and determined its role in articular cartilage homeostasis. Cbfβ significantly decreased in aged mouse articular cartilage and human OA cartilage. Articular chondrocyte-specific *Cbfb*-deficient mice (*Cbfb*^△*ac/*△*ac*^) exhibited early cartilage degeneration at 20 weeks of age and developed OA at 12 months. *Cbfb*^△*ac/*△*ac*^ mice showed enhanced OA progression under the surgically induced OA model in mice. Mechanistically, forced expression of Cbfβ rescued Type II collagen (Col2α1) and Runx1 expression in Cbfβ-deficient chondrocytes. TGF-β1-mediated Col2α1 expression failed despite the p-Smad3 activation under TGF-β1 treatment in Cbfβ-deficient chondrocytes. Cbfβ protected Runx1 from proteasomal degradation through Cbfβ/Runx1 complex formation. These results indicate that Cbfβ is a novel anabolic regulator for cartilage homeostasis, suggesting that Cbfβ could protect OA development by maintaining the integrity of the TGF-β signaling pathway in articular cartilage.

## 1. Introduction

Osteoarthritis (OA), the most common and painful disease of the joints, affects nearly half of the world’s elderly population and represents an enormous socioeconomic challenge [1,2]. OA is a joint disease with structural and functional changes in the articular cartilage, subchondral bone, ligaments, capsule, synovium, sensory nerve endings, meniscus, and periarticular muscles. Among these structural components, many studies focus on cartilage or subchondral bone. With regard to OA initiation, both subchondral bone and articular cartilage changes are critical factors for OA initiation [1,3,4,5]. A dynamic equilibrium between synthesis and degradation of the extracellular matrix, such as Col2α1, proteoglycans, and Mmp13, controls articular cartilage homeostasis and integrity [4]. The inactivation of one copy of Col2α1 leads to articular cartilage degeneration and increased OA development [6]. In the early stage of OA, cartilage matrix degradation occurs in the superficial zone of the cartilage but later extends to deeper zones as OA progresses [7]. Mmp13 is a major enzyme that targets the breakdown of cartilage extracellular matrix such as Type II collagen and proteoglycans [8]. A clinical investigation revealed a strong association between articular cartilage destruction and high Mmp13 expression [9].

The tight control of TGF-β signaling plays an essential role in articular cartilage homeostasis [10]. The injection of TGF-β into the knee joint increases proteoglycan, whereas the injection of recombinant soluble TGF-β type II receptor, an endogenous inhibitor, markedly worsens OA phenotypes [11,12]. TGF-β Type I receptor ALK1 (Activin Receptor-like Kinase) correlates well with Mmp13 expression, whereas ALK5 correlates with aggrecan and collagen type II. In senescent and OA articular chondrocytes, ALK5 expression is significantly decreased compared to ALK1, increasing the ALK1/ALK5 ratio, which is associated with the upregulation of Mmp13 in OA [13]. Indeed, the chondrocyte-specific loss of Smad3, ALK5, or the overexpression of dominant-negative TGF-β type II receptor in mice results in OA development at an early stage [14,15]. These findings indicate that abnormal TGF-β signaling increases OA development. Thus, the integrity of the TGF-β signaling pathway is essential for the healthy maintenance of articular cartilage and for preventing OA. Here, we explore the TGF-β signaling integrity regulated by Cbfβ.

Cbfβ is a partner protein of the runt-related transcription factor (Runx) family, which includes Runx1, Runx2, and Runx3. It interacts with the Runt domain of the Runx family to enhance its DNA binding properties and transcription activity [16,17]. Runx1 is a pivotal transcription factor for chondrogenesis, chondrocyte proliferation, and survival. Runx1 in mouse embryonic mesenchymal cells results in a potent induction of early chondrocyte differentiation markers, such as Type II collagen, but not the hypertrophy marker, type X collagen [18]. Runx1 also acts on the superficial zone of articular chondrocytes to stimulate chondrocyte proliferation and promote anabolic protein expression to maintain articular cartilage integrity [19]. Deletion or carboxyl terminus truncation of Runx2 inhibits endochondral bone formation with the arrest of chondrocyte maturation, resulting in the complete absence of mature osteoblast formation [20,21,22]. Runx2 and Runx3 are both expressed in cartilaginous condensations of the mouse limb. The cooperative roles of these two proteins in mediating chondrocyte maturation, hypertrophy, and cartilage formation have been established in vivo using double-knockout mice [23]. The homozygous deletion of *Cbfb* in mice causes embryonic lethality during pregnancy due to brain hemorrhage and reduces Runx1 stability and transcriptional activity to block fetal hepatic hematopoiesis [24,25]. Cbfβ plays an essential role in endochondral bone formation through skeletal tissue cell-type-specific *Cbfb* deletions, including mesenchymal cells [26,27], chondrocytes [28,29], and osteoblasts [30]. Since articular chondrocytes remain as non-hypertrophied hyaline cartilage during development, it is interesting to determine how the joint degenerates and progresses to OA as the articular chondrocytes die prematurely or undergo abnormal hypertrophy [7]. However, the role of Cbfβ during articular cartilage development and the evolution of OA is not well understood. 

To determine the function of Cbfβ in articular cartilage and its effect on the development of OA, we analyzed the results of the deletion of *Cbfb* from articular chondrocytes using *Gdf5-Cre* and *Cbfb-floxed* mice together with a surgically induced OA model.

## 2. Materials and Methods

### 2.1. Antibodies and Reagents

Antibodies and reagents are listed in Table 1.

### 2.2. Mice and Experimental OA

Mice harboring an articular chondrocyte-specific deletion of *Cbfb* (*Cbfb*^Δ*ac/*Δ*ac*^) were established by crossing *Gdf5-Cre* transgenic mice (*Cre^Tg/+^*, kindly provided by Dr. David M Kingsley from Stanford University, Stanford, CA, USA) [31] with *Cbfb^fl/fl^* mice [32], both maintained on a C57BL/6N background. The loxP sites of the *Cbfb^fl/fl^* mice encompass exon 5 of Cbfβ, resulting in articular chondrocyte-specific deletion of exon 5 in *Cbfb*^△*ac/*△*ac*^ mice. *Cbfb^fl/fl^* served as wild-type control (WT). In twelve-week-old male mice, the destabilization of the medial meniscus (DMM) surgery was conducted in both knees, and mice were sacrificed at 8 weeks after DMM surgery to assess OA progression (each group *n* = 6) [33]. We used twelve-month-old male mice for spontaneous OA progression analysis (WT = 5, *Cbfb*^△*ac/*△*ac*^ = 5). All animal procedures followed the guidelines issued by the Institutional Animal Care and Use Committee of Kyungpook National University (KNU-201455).

### 2.3. Human Subjects

Human OA articular cartilage tissues were obtained from OA surgery patients through total knee arthroplasty. The Institutional Review Board of Kyungpook National University Hospital approved using human OA cartilage. We obtained written informed consent from all patients prior to the surgical procedure (IRB File No of KNUH 2022-01-010-001). The human sample list is described in Table 2.

### 2.4. β-Galactosidase (β-gal) Staining

Whole tissues were fixed for 30 min in 4% paraformaldehyde (PFA) in 0.1 M potassium phosphate buffer (pH 7.4) containing 2 mM MgCl_2_, 5 mM ethylene glycol tetra-acetic acid (EGTA), then washed for 30 min with 0.1 M potassium phosphate buffer. Tissues were kept in the X-gal staining solution overnight at 37 °C and then washed briefly in phosphate-buffered saline (PBS). We made cryosections of the stained samples, fixed the slides briefly in 4% PFA, washed them in PBS, and counterstained them with Nuclear Fast Red (Sigma-Aldrich, St. Louis, MO, USA). Samples were washed in PBS and analyzed with a Leica microscope [34].

### 2.5. Assessment of OA Severity

Knee joints were fixed with 4% PFA and decalcified with 10% ethylenediaminetetraacetic acid (EDTA; pH 7.4) for 4 weeks. Decalcified tissues were dehydrated by ethanol, embedded in paraffin, and then sectioned with a thickness of 3 μm. We performed Safranin-O staining as in the previous study [5]. Briefly, sections were treated in the following order: deparaffinization, rehydration, and soaking in Weigert’s iron hematoxylin solution for 10 min each, followed by fast green solution and 0.1% Safranin-O solution for 5 min each. Assigning OA progression followed the Osteoarthritis Research Society International (OARSI) diagnosis criteria [5].

### 2.6. Immunohistochemistry

We quenched the sections with 3% H_2_O_2_ and retrieved antigens by boiling them in TEG buffer (1.211 g of Tris and 0.190 g of EGTA in 1 L of MilliQ-water, pH 9.0). After blocking with 1% bovine serum albumin for 1 h at room temperature, sections were incubated with anti-Cbfβ, anti-Runx1, anti-Type II collagen, and anti-Mmp13 antibodies overnight at 4 °C and incubated with goat anti-rabbit IgG or goat anti-mouse IgG conjugated HRP for 1 h. Signals developed with a DAB substrate-chromogen system (Dako, Carpinteria, CA, USA).

### 2.7. Cell Culture

Primary articular chondrocytes were derived from the hindlimb knee joint of wild-type and *Cbfβ*^△*ac/*△*ac*^ mice on postnatal day 5, as previously described [35]. Chondrocytes were grown in Dulbecco’s modified Eagle’s medium (DMEM) (Lonza, ME, USA) containing 10% fetal bovine serum (FBS) (Gibco-BRL, Gaithersburg, MD, USA) and the appropriate penicillin/streptomycin. Chondrogenic ATDC5 cells were cultured in DMEM/F12 (Lonza) medium supplemented with 5% FBS, 10 μg/mL human transferrin, and 3 × 10^−8^ M sodium selenite. Chondrocytes were plated at a density of 2 × 10^5^ cells/well on 6-well plates for drug treatment. Protein or total RNA was isolated from chondrocytes after 24 h treatment with TGF-β1 (1, 10, or 100 ng/mL), IL-1β (10 ng/mL), or TNF-α (10 ng/mL).

### 2.8. Western Blot Analyses

Total protein lysates were isolated from human OA articular cartilage, mouse tibia articular cartilage of 12-month-old mice, and articular chondrocytes derived from articular cartilage by M-PER™ Mammalian Protein Extraction Reagent (Thermo Fisher Scientific, Rockford, IL, USA) containing a cocktail of protease inhibitors and phosphatase inhibitors. Proteins were separated on 10% SDS-polyacrylamide gels and transferred onto polyvinylidene difluoride (PVDF) membranes. Western blots were performed using the indicated antibodies [36].

### 2.9. qRT-PCR Analyses

Total RNAs were isolated from hind-limb articular cartilage of 12-month-old mice or primary cultured articular chondrocytes using the easy-BLUE Total RNA Extraction Kit (iNtRON Biotechnology, Seongnam-si, Gyeonggi-do, Korea) and cDNA was synthesized from 1 μg of total RNA using SuperScript II Reverse Transcriptase (Invitrogen, CA, USA). qRT-PCR was performed using the Power SYBR green master mixture (Applied Biosystems, Foster, CA, USA). The qRT-PCR primers were designed using Primer Express software (Applied Biosystems). Sequences of qRT-PCR primers are shown in Table 3.

### 2.10. Co-Immunoprecipitation (Co-IP) Assay

Primary articular chondrocytes were plated at a density of 2 × 10^6^ cells/well in 100 mm plates for transfection experiments. Cells were transfected with 10 μg of DNA, including 5 μg of the pcDNA3.1-myc-Cbfβ and 5 μg of pCS4-2Flag-Runx1 by using Lipofectamine 2000 (Invitrogen) and grown for 24 h. Cell lysates (400 μg) were bound with anti-Myc at 4 °C overnight with gentle agitation, then incubated with 0.1 g of Sepharose A beads (GE Healthcare, Chicago, IL, USA) in lysis buffer for 2 h at 4 °C with gentle agitation. Mixtures were washed with lysis buffer five times, and proteins were eluted by boiling with 50 μL of protein loading buffer. Co-IP was performed using Western blotting with anti-Myc or anti-Flag antibody [37].

### 2.11. Poly-Ubiquitination Assay

ATDC5 chondrocytes were transfected with 12 μg of DNA, including 4 μg of pCS4-Flag-Runx1, 4 μg of pcDNA3.1-HA-Ubiquitin, and 4 μg of pCS4-Myc-Cbfβ by using Lipofectamine 2000 (Invitrogen, Waltham, MA, USA) and grown for 24 h. After 24 h from transfection, cells were incubated with 20 μm of MG132 for 2 h. Cell lysates (400 μg) were allowed to bind with anti-Flag antibody at 4 °C overnight with gentle agitation, then incubated with 0.1 g of Sepharose A beads (GE Healthcare, Chicago, IL, USA) in lysis buffer for 2 h at 4 °C with gentle agitation. Proteins were eluted by boiling with 50 μL of protein loading buffer. Ubiquitination levels were evaluated with a Western blot with anti-HA antibody [37].

### 2.12. Statistical Analyses

We used GraphPad Prism 9 (GraphPad, San Diego, CA, USA) for statistical analysis. A paired or unpaired Student’s t-test was used to analyze statistically significant comparisons between the two groups. We considered *p* < 0.05 to be statistically significant (* *p* < 0.05, ** *p* < 0.01), and ns represents no statistical significance. Data are expressed as mean ± SD or mean ± SE.

## 3. Results

### 3.1. Cbfβ was Enhanced by Anabolism and Suppressed by Catabolism

To assess the involvement of Cbfβ in articular cartilage metabolism, we analyzed the expression of Cbfβ under TGF-β1, interleukin-1β (IL-1β), or TNF-α activation. Treatment with various concentrations (1, 10, and 100 ng/mL) of TGF-β1, one of the pivotal anabolic factors in articular cartilage, increased p-Smad3 and Cbfβ protein levels in the articular chondrocytes (Figure 1a). TGF-β1 also increased the mRNA expression of Cbfβ, Type II collagen, and Aggrecan (Figure 1b). The pro-inflammatory cytokines IL-1β or TNF-α decreased Cbfβ expression along with Type II collagen and Aggrecan, while Mmp13 expression increased in primary cultured mouse articular chondrocytes (Figure 1c). We confirmed that Cbfβ decreased under IL-1β or TNF-α treatment in the fluorescent staining (Figure 1d). Moreover, Cbfβ silencing in chondrocytes upregulated the expression of Mmps such as Mmp9, Mmp13, Mmp14, and Mmp15, and inflammatory cytokines IL-6, IL-17, IL-18, and IL-22 in the chondrocytes (Appendix A). Taken together, Cbfβ expression was enhanced by anabolism and suppressed by catabolism in the articular chondrocytes.

### 3.2. Cbfβ Loss Is Involved in Articular Cartilage Degeneration

We then investigated whether there was a correlation between Cbfβ expression and articular cartilage degeneration in vivo. Aged articular cartilage reduced Cbfβ and its partner protein Runx1, but the expression of Runx2 and Runx3 was not significantly different compared to the young articular cartilage (Figure 1e). Moreover, the expression of Cbfβ decreased in mouse (Figure 1f) and human osteoarthritic cartilage (Figure 1g). These results suggest that a loss of Cbfβ may be involved in articular cartilage degeneration. Next, we investigated Cbfβ functions in joint development using articular-cartilage-specific *Cbfb* deleted *Gdf5-Cre; Cbfβ^fl/fl^* (*Cbfb*^△*ac/*△*ac*^) mice. With *Rosa26* reporter (*R26R*), mice *Gdf5-Cre* activities were explicitly observed in the articular cartilage through β-galactosidase staining (Appendix A), with around 75% deletion in articular cartilage by qRT- PCR (Appendix A) and immunofluorescent staining in E16.5 (Appendix A) and 20-week-old (Appendix A). *Cbfb*^△*ac/*△*ac*^ mice displayed typical joint and cartilage formation at E14.5 and E16.5 embryonic stages (Figure 2a,b). To further understand the effect of Cbfβ on articular cartilage homeostasis, we performed an articular cartilage integrity assessment with the histochemical analysis of 12-month-old mouse joints. *Cbfb*^△*ac/+*^ mice displayed higher OARSI scores than WT mice but without statistical significance. However, *Cbfb*^△*ac/*△*ac*^ mice exhibited high OARSI scores with severe cartilage destruction (Figure 2c,d). Cbfβ, Aggrecan, and Col2α1 mRNAs decreased, whereas Mmp13 expression increased in the articular cartilage of 12-month-old *Cbfb*^△*ac/*△*ac*^ mice (Figure 2e). These results indicate that Cbfβ physiologically protects articular cartilage during aging and that the deletion of *Cbfb* causes age-dependent spontaneous OA progression.

### 3.3. Genetic Deletion of Cbfb Accelerated OA Progression

Cbfβ deficiency induced spontaneous OA development and presumably accelerated knee joint degradation in a surgically induced OA model. To explore this idea using a DMM surgery-induced mouse OA model, we confirmed the *Cbfb*^△*ac/*△*ac*^ mouse joints’ OA progression. Safranin O staining and scoring using the OARSI method revealed that *Cbfb*^△*ac/*△*ac*^ mouse joints induced severe damage to articular cartilage compared with controls. Indeed, *Cbfb*^△*ac/*△*ac*^ mouse joints, even the sham group, showed more cartilage degeneration with a higher OARSI score than WT mice at 20 weeks (Figure 3a,b). Immunohistochemical staining showed reduced articular cartilage proteins that provide protection, such as Runx1 and Col2α1 (Figure 3c). Conversely, the matrix-degrading enzyme Mmp13 increased on the articular surface of *Cbfb*^△*ac/*△*ac*^ mice compared to those of WT mice (Figure 3c). TGF-β is one of the most critical anabolic regulators for articular cartilage integrity. Hence, we determined whether Cbfβ could modulate TGF-β signaling in the *Cbfb*^△*ac/*△*ac*^ mouse articular cartilage. In fluorescence staining, the signal of p-Smad3 unexpectedly increased in the *Cbfb*^△*ac/*△*ac*^ mouse articular cartilage (Figure 3d,e). These data raised an additional question as to why TGF-β signaling activation could not increase Type II collagen expression in *Cbfb*^△*ac/*△*ac*^ mouse articular cartilage.

### 3.4. Cbfβ Modulates Articular Cartilage Integrity by Modulating TGF-β Signaling

To clarify how Cbfβ is involved in the TGF-β1-mediated Type II collagen expression, we performed a TGF-β1-responsive assay in the Cbfβ-deleted chondrocytes. Interestingly, TGF-β1 treatment increased p-Smad3 in the *Cbfb*^△*ac/*△*ac*^ articular chondrocytes compared to *Cbfβ^fl/fl^* cells (Figure 4a). Consistent with in vivo data, TGF-β1-mediated Cbfβ and Col2α1 (Type II collagen) expression were reduced in *Cbfb*^△*ac/*△*ac*^ articular chondrocytes compared with *Cbfβ^fl/fl^* (Figure 4b,c). These results indicate that Cbfβ plays an essential role in maintaining TGF-β signaling pathway integrity in the articular cartilage.

### 3.5. Cbfβ Stabilizes Runx1 in Articular Chondrocytes

Runx1, Runx2, and Runx3 are crucial targets of TGF-β superfamily signaling [38]. Runx1 is a central regulator of articular cartilage integrity by coordinating YAP, TGF-β, and Wnt signaling in articular cartilage formation and maintenance by enhancing Type II collagen and matrix production in collaboration with Sox trios [39]. We, therefore, hypothesized that Runx1 proteolysis resulted in accelerated cartilage degeneration in the Cbfβ-deleted articular chondrocytes [40]. The levels of Runx1 and Col2α1 were reduced in Cbfβ-deleted articular chondrocytes from *Cbfb^△ac/^^△ac^* mice, whereas the rescue of Cbfβ in *Cbfb*-deleted articular chondrocytes restored Runx1 and Col2α1 (Figure 5a,b). To further investigate the effects of Cbfβ on Runx1 stability in articular chondrocytes, we performed co-immunoprecipitation and poly-ubiquitination (Figure 5c). In articular chondrocytes, Cbfβ can endogenously bind Runx1 and protect Runx1 from polyubiquitination-mediated proteasomal degradation (Figure 5c,d). These results suggest that Cbfβ makes a complex with Runx1, stabilizing Runx1 in articular chondrocytes.

## 4. Discussion

In this study, we found that the articular-cartilage-specific deletion of Cbfβ resulted in spontaneous OA development and exacerbated OA progression in the surgically induced DMM model. The expression of Cbfβ decreased with aging and OA progression in articular cartilage, and only Runx1 among the Runx transcription factors was downregulated in response to a decrease in Cbfβ (Figure 1c,d). The absence or rescue of Cbfβ in *Cbfb*-deleted articular chondrocytes determined the levels of Runx1 and the Cbfβ/Runx1 complex as well as Col2α1, indicating that Cbfβ/Runx1 complex formation plays a crucial role in maintaining articular cartilage homeostasis. Moreover, Cbfβ as a component of the Cbfβ/Runx1 complex was a key regulator of TGF-β signaling, with the modulation of p-Smad3 in *Cbfb*-deleted articular chondrocytes. Proper Cbfβ/Runx1 complex formation via TGF-β signaling is vital for maintaining normal articular chondrocyte functional integrity and preventing degenerative cartilage diseases of joints such as OA.

The physiological maintenance of articular cartilage requires tight control of the TGF-β signaling pathway [10]. Our study demonstrates the critical role of Cbfβ and Runx1 as a mediator of TGF-β signaling (Figure 5e). First, Cbfβ and Runx1, like various components of the TGF-β signaling pathway, such as the ALK1/ALK5 ratio [13], decreased with aging and OA progression. The activation of TGF-β signaling increased Cbfβ and Runx1 expression and Cbfβ/Runx1 complex formation. The deletion of Cbfβ attenuated the activation of the TGF-β signaling pathway; however, the forced expression of Cbfβ recovered Runx1 and Col2α1 (Type II collagen) expression in Cbfβ-deficient chondrocytes. Moreover, the deletion of Cbfβ increased p-Smad3 in the articular cartilage of *Cbfb*^△*ac/*△*ac*^ mice. The increase in p-Smad3 may result from the regulatory role of the negative feedback loop in the TGF-β signaling pathway. Unknown factors regulated by or involved in the Cbfβ/Runx1 complex in articular chondrocytes need further studies.

Several studies have suggested the contributions of Runx transcription factors to OA. For example, Runx1 contributes to articular cartilage maintenance by increasing cartilage matrix production and inhibiting hypertrophic differentiation [40,41]. Another study suggests that the small-molecule KGN binds to filamin A, a cytoplasmic sequestrant of Cbfβ, releasing Cbfβ from filamin A, which translocates Cbfβ to the nucleus and binds to Runx1 but not Runx2 of articular chondrocytes [42]. Runx1 is known to stimulate the differentiation of mesenchymal cells into chondrocytes by regulating the expression of Type II collagen rather than type X collagen, a hypertrophy marker [18,42]. This study showed that Cbfβ levels correlated well with Runx1 but not with Runx2 and Runx3. In aged articular cartilage, Cbfβ and Runx1 decreased, but Runx2 and Runx3 did not, which also correlated well with the articular cartilage phenotype of *Cbfb*^△*ac/*△*ac*^ mice. Our studies revealed that disruption of TGF-β signaling by deletion of Cbfβ in articular chondrocytes increased catabolic cytokines and enzymes such as IL-6, IL-17, IL-18, IL-22, Mmp 9, Mmp 13, Mmp14, and Mmp15. Therefore, the positive relationship of the Cbfβ/Runx1 complex with Type II collagen is essential to the TGF-β signaling pathway in articular cartilage integrity. 

Regarding Runx2 and Runx3, Runx2 is negligibly expressed, while Runx3 is comparable to Runx1 in articular cartilage. Our study showed that diminished Runx1 correlated well with the level of Cbfβ, whereas increased Runx2 and Runx3 in the human OA samples did not. Our previous studies showed the protection of Runx2 by Cbfβ in both chondrocytes and osteoblasts [29,30], suggesting the presence of differential heterodimerization of Cbfβ with Runx transcription factors in skeletal tissues. Some cases of OA upregulate Runx2 with a transition from articular chondrocytes to hypertrophic chondrocytes [43]. Runx2 haploinsufficient mice show delayed OA progression after the induction of knee joint instability [44]. Additionally, the deletion of Runx2 in articular chondrocytes slows the progression of surgery-induced OA [45]. Chondrocyte-specific Runx2 overexpression accelerates OA progression [46]. These studies indicate a positive relationship between OA progression and aberrant expression of Runx2 in articular chondrocytes. However, the Cbfβ investigation showed that Runx1, but not Runx2 or Runx3, was primarily involved in the aggravated OA phenotype in the DMM model and Cbfβ-deleted spontaneous OA and human OA samples. Thus, the local upregulation of Cbfβ expression using various methods, including small molecules, AAV vectors, or other mRNA gene delivery systems in joints, might play a critical role in protecting and delaying cartilage degeneration, which is expected to provide a therapeutic method for OA treatment [47].

This study has a few limitations. How the deletion of Cbfβ increased p-Smad3 in the articular cartilage of *Cbfb*^△*ac/*△*ac*^ mice in vivo and primary chondrocytes in vitro is still unclear. P-Smad3 upregulation may result from the negative feedback loop in the TGF-β signaling pathway or a compensatory mechanism. Unknown factors regulated by or involved in the Cbfβ/Runx1 complex in articular chondrocytes need further studies. 

## 5. Conclusions

In this study, the deletion of Cbfβ in articular cartilage decreased Runx1 and Type II collagen, which enhanced OA progression spontaneously in *Cbfb*^△*ac/*△*ac*^ mice. These results indicate that Cbfβ regulates Runx1 protein stability and subsequently modulates Runx1 target genes, such as Type II collagen, in the articular cartilage. Our findings indicate that Cbfβ prevents articular cartilage destruction by maintaining the integrity of the TGF-β signaling pathway through the stabilization of Runx1.

## Figures and Tables

**Figure 1 cells-12-01064-f001:**
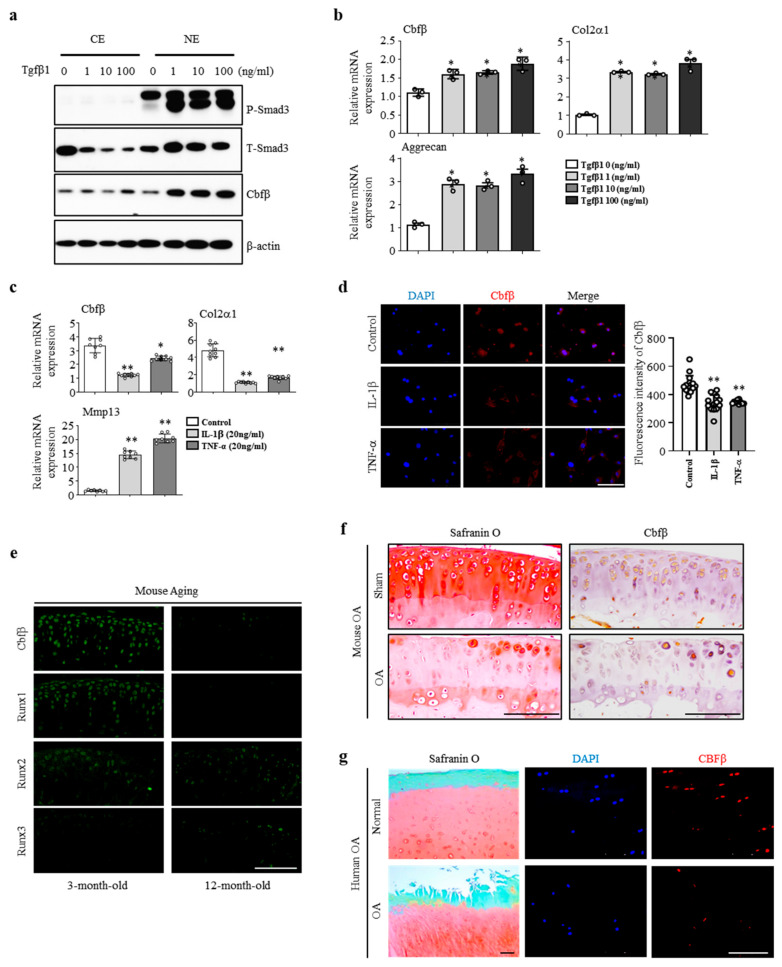
Cbfβ loss is involved in OA pathogenesis. (**a**) Cbfβ and pSmad 3 protein expression detected using Western blotting in articular chondrocytes with TGF-β1 treatment. CE, cytoplasmic extracts; NE, nuclear extracts; (**b**) Cbfβ, Col2α1, and Aggrecan mRNA expression detected by qRT-PCR in articular chondrocytes with TGF-β1 treatment. * *p* < 0.05 vs. TGF-β1 0 (ng/ml). (**c**) Cbfβ, Cols2α1, and Mmp13 mRNA expression detected by qRT-PCR in articular chondrocytes with IL-1β or TNF-α treatment. * *p* < 0.05, ** *p* < 0.01 vs. control. (**d**) Cbfβ detection by immunofluorescence staining after treatment of IL-1β and TNF-α in articular chondrocytes. Scale bars, 100 μm. (**e**) Expression of Cbfβ, Runx1, Runx2, and Runx3 at 3 and 12 months of age was evaluated with immunofluorescence staining (*n* = 3). Scale bars, 100 μm. (**f**) Mouse joint OA development was assessed with Safranin O staining and Cbfβ expression was confirmed by immunohistochemical staining. Scale bars, 100 μm. (**g**) Human OA development was assessed with Safranin O staining. DAPI staining, and Cbfβ expression was conducted with immunofluorescent staining (*n* = 6). Scale bars, 100 μm.

**Figure 2 cells-12-01064-f002:**
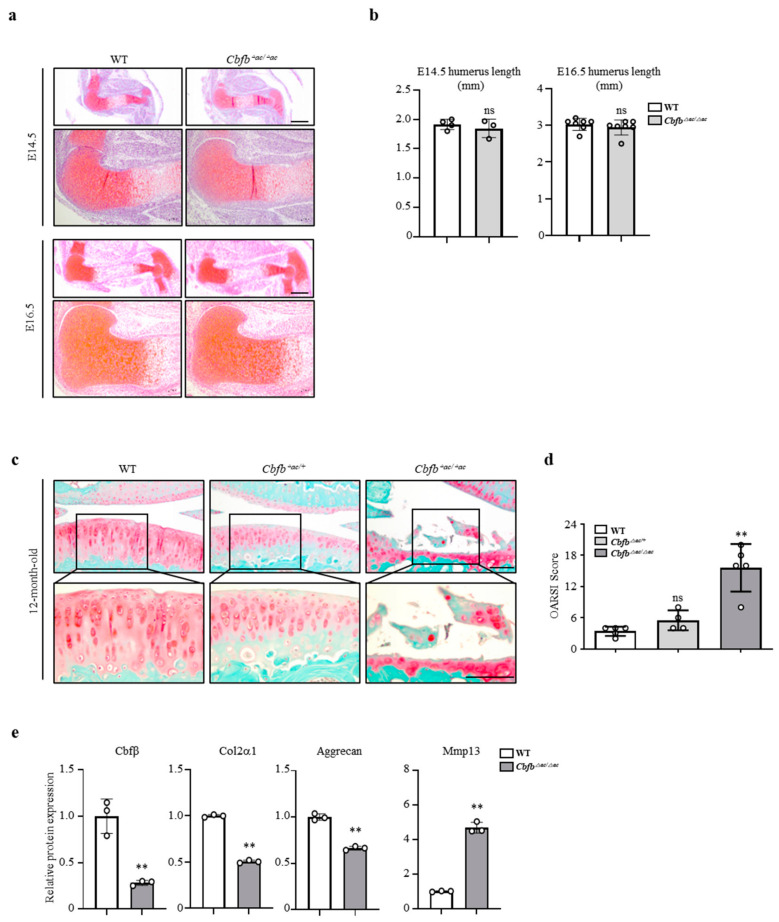
Articular-cartilage-specific Cbfβ-deleted mice exhibit spontaneous cartilage degeneration. (**a**,**b**) Evaluation of joint and limb formation via Safranin-O staining at E14.5 (WT = 4, *Cbfb*^∆*ac/*∆*ac*^ =3) and E16.5 (WT = 7, *Cbfb*^∆*ac/*∆*ac*^ = 7) of *Cbfb*^∆*ac/*∆*ac*^ mice. The humerus length was analyzed with the Leica program. Scale bars, 500 μm and 100 μm. (**c**) Spontaneous OA development assessed via Safranin O staining in 12-month-old *Cbfb*^∆*ac/*∆*ac*^ mice. (**d**) OA severity was determined by histological analysis according to OARSI guidelines. Scale bars, 100 μm (WT = 5, *Cbfb*^∆*ac/*∆*ac*^ = 4, *Cbfb*^∆*ac/*∆*ac*^ = 5). ** *p* < 0.01 vs. WT. (**e**) Cbfβ, Col2α1, Aggrecan, and Mmp13 mRNA expression were evaluated using qRT-PCR in 12-month-old *Cbfb^∆ac/∆ac^* articular cartilage. ** *p* < 0.01 vs. WT, ^ns^ not significant.

**Figure 3 cells-12-01064-f003:**
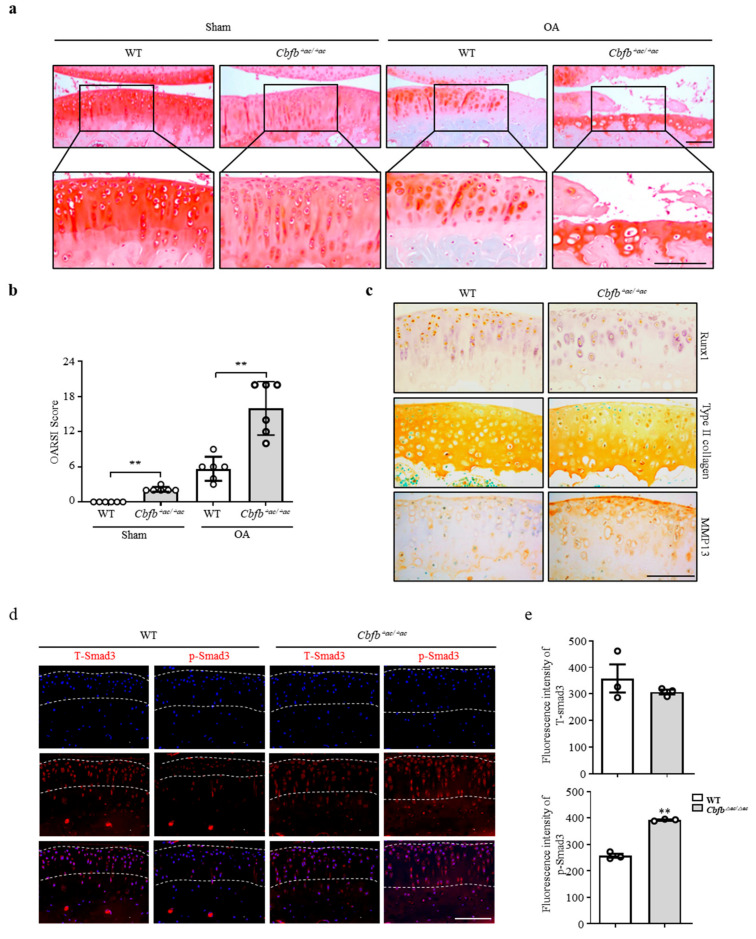
Deletion of *Cbfb* in articular chondrocytes accelerated OA pathogenesis progression. (**a**) Articular cartilage was severely destroyed in *Cbfβ*^△*ac/*△*ac*^ mice; the destruction was assessed using Safranin O staining 8 weeks after DMM surgery. (**b**) The severity of OA was analyzed according to OARSI guidelines. ** *p* < 0.01 (*n* = 6). Scale bars, 100 μm. (**c**) Immunohistochemical staining was performed on paraffin sections using anti-Runx1, -Col2α1, and -Mmp13 antibodies. (**d**) Expression of total Smad3 (T-Smad, left panels) and p-Smad3 (right panels) in articular cartilage was assessed using immunofluorescence staining in WT and *Cbfβ*^△*ac/*△*ac*^ 20-week-old mice. (**e**) The T-Smad3 and p-Smad3 fluorescence intensities were analyzed with the Leica fluorescence analysis program. ** *p* < 0.01 WT (*n* = 3). Scale bars, 100 μm.

**Figure 4 cells-12-01064-f004:**
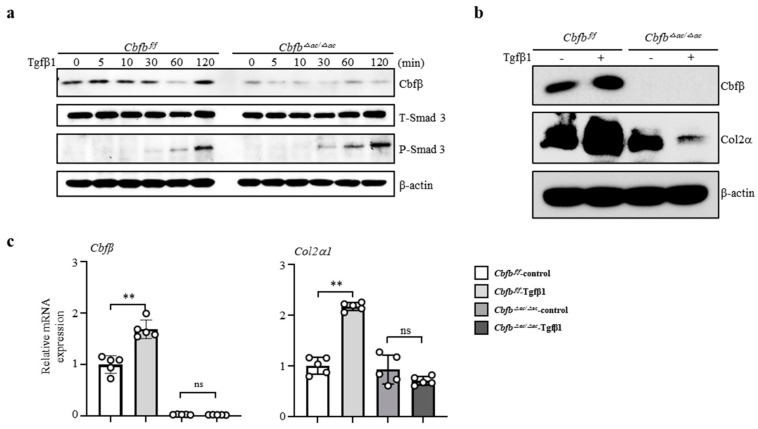
Impaired TGF-β1-mediated Col2α1 transcription in Cbfβ-deficient articular chondrocytes. (**a**) Cbfβ, p-Smad3, and T-smad3 expression were evaluated with Western blotting in *Cbfβ^△ac/^^△ac^* articular chondrocytes. (**b**) Effects of TGF-β1 treatment for 24 h on Cbfβ and Col2α1 (Type II collagen) protein levels using Western blotting. (**c**) The expression of Cbfβ and Col2α1 mRNA was measured by qRT-PCR. ** *p* < 0.01, ^ns^ not significant.

**Figure 5 cells-12-01064-f005:**
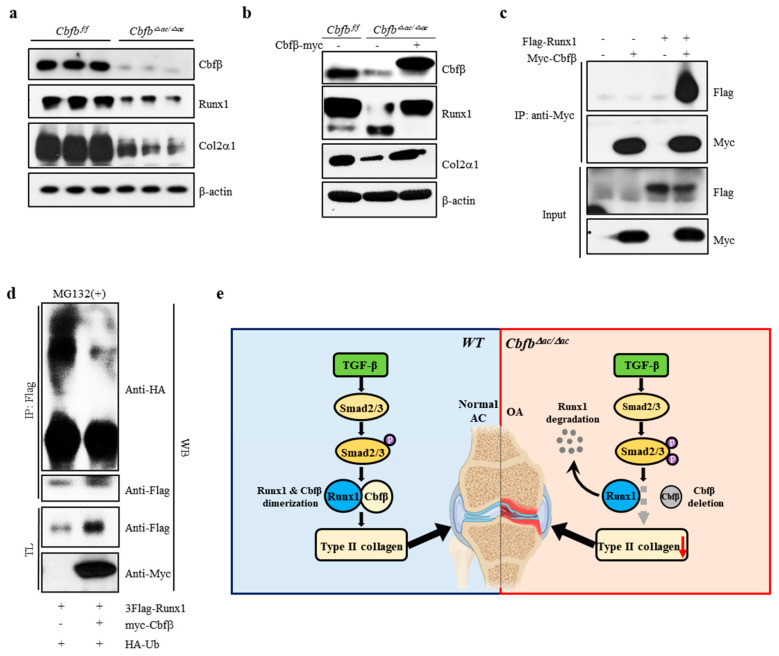
Requirement of Cbfβ for Runx1 stabilization in articular chondrocytes. (**a**) Total protein was extracted from primary articular chondrocytes derived from WT or *Cbfβ*^△*ac/*△*ac*^ mice, and Western blotting was performed with anti-Cbfβ, -Runx1, -Col2α1, and -β-actin antibodies (β-actin as the loading control. (**b**) Primary cultured articular chondrocytes derived from *Cbfβ*^△*ac/*△*ac*^ mice were transiently transfected with Myc-Cbfβ for 24 h, and the expression levels of Cbfβ, Runx1, Col2α1, Mmp13, and β-actin were analyzed using Western blotting (WT was relative control). (**c**) Primary cultures of articular chondrocytes were co-transfected with vectors expressing Flag-Runx1 and Myc-Cbfβ for 24 h. Total extracts were immunoprecipitated with anti-Myc antibody, and Runx1 was detected with anti-Flag antibody using Western blotting. (**d**) For polyubiquitination assay, ATDC5 chondrocytes were transfected with HA-Ub, Flag-Runx1, and Myc-Cbfβ for 24 h and treated with 10 μM MG132 for the last 2 h. After immunoprecipitation of Runx1 using anti-Flag antibody, Ub, Runx1, and Cbfβ levels were determined with Western blotting (WB) using anti-HA, anti-Flag, and anti-Myc antibodies. TL, total lysates. (**e**) Schematic diagram of how the Cbfβ/Runx1 complex affects TGF-β signaling to regulate articular cartilage homeostasis. Deletion of Cbfβ induces spontaneous OA and surgically induced accelerated OA due to rapid Runx1 degradation.

**Table 1 cells-12-01064-t001:** Antibodies and Reagents.

Reagent Type	Designation	Identifiers
Antibody	Anti-p-Smad3 (rabbit polyclonal)	Cell Signaling Technology #9520
Antibody	Anti-smad3 (rabbit polyclonal)	Cell Signaling Technology #9513
Antibody	Anti-Cbfβ (rabbit polyclonal)	Santa Cruz #20693
Antibody	Anti-β-actin (mouse monoclonal)	Santa Cruz #47778
Antibody	Anti-Col2α1 (rabbit polyclonal)	Abcam #53047
Antibody	Anti-Runx1 (mouse monoclonal)	Santa Cruz #365644
Antibody	Anti-Myc-Tag (rabbit polyclonal)	Abcam #9106
Antibody	Anti-Myc-Tag (mouse monoclonal)	Invitrogen #P/N46-0603
Antibody	Anti-Flag-Tag (mouse monoclonal)	Sigma #F1804
Antibody	Anti-Mmp13 (mouse monoclonal)	Calbiochem #IM-78
Protein ladder	Protein marker	Dokdo-1032
Chemical compound, drug	Type IV collagenase	Gibco # 9001-12-1
Chemical compound, drug	Lipofectamine 2000	Invitrogen #11668019
Chemical compound, drug	Hematoxylin	Sigma-Aldrich #H9627
Chemical compound, drug	Fast green	Sigma-Aldrich #F7258
Chemical compound, drug	Ferric chloride	Junsei #18510S0301
Chemical compound, drug	Safranin O	Sigma-Aldrich #S-2255
Chemical compound, drug	Mammalian cell lysis buffer	Thermo Fisher Scientific #78501
Chemical compound, drug	Easy-Blue solution	iNtRON #17061
Chemical compound, drug	DAB substrate-chromogen system	Dakocytomation #K3468
Chemical compound, drug	Bovine Serum Albumin	Sigma-Aldrich #A9418
Chemical compound, drug	4% Paraformaldehyde	Biosolution #BP031
Chemical compound, drug	Mayor’s Hematoxylin	Muto pure chemicals #3000-2
Chemical compound, drug	Eosin	Muto pure chemicals #3200-2
Chemical compound, drug	Ethylenediaminetetraacetic acid	Junsei #17388-0401
Recombinant protein	IL-1β	R&D Systems #401-ML
Recombinant protein	TNF-α	R&D Systems #410-MT
Recombinant protein	Tgf-β1	R&D Systems #7666-MB

**Table 2 cells-12-01064-t002:** Human Samples.

No.	Age	Gender	ICRS Grade	Joint	Weight (kg)	Height (cm)	RA	Others
1	65	F	4	Knee	66	158	X	heart valve disease
2	76	F	4	Knee	71	155	X	HTN, DM, unstable angina, asthma
3	64	M	3	Knee	67	168	X	HTM
4	83	F	4	Knee	55	151	X	HTN
5	76	F	3	Knee	55	155	X	HTN, DM
6	71	F	4	Knee	70	159	X	HTM, DM, AF

F, Female. M, Male. ICRS, International Cartilage Repair Society. RA, rheumatoid arthritis. HTN, hypertension. DM, diabetes mellitus. AF, Atrial fibrillation.

**Table 3 cells-12-01064-t003:** Primer sequences for qRT-PCR.

Target Gene	Forward Sequence	Reverse Sequence
*Gapdh*	GCATCTCCCTCACAATTTCCA	GTGCAGCGAACTTTATTGATGG
*Cbfb*	TATGGGTTGCCTGGAGTT TG	AAGGCCTGTTGTGCTAATGC
*Col2α1*	TTCCACTTCAGCTATGGCGA	GACGTTAGCGGTGTTGGGAG
*Aggrecan*	GAGAGAGGCGAATCGAACGA	CGTGAAGGGCAGCTGGTAAT
*Mmp9*	AAACCAGACCCCAGACTCCTC	GAGGACACAGTCTGACCTGAA
*Mmp13*	GCCAGAACTTCCCAACCATG	TCAGAGCCCAGAATTTTCTCC
*Mmp14*	GGATGGACACAGAGAACTTCGTG	CGAGAGGTAGTTCTGGGTTGAG
*Mmp15*	CTGAGCAGCTATGGCACAGACA	TGCTGTGTCTCCTCGTTGAAGC
*IL-6*	TTGCCTTCTTGGGACTGATG	CTGAAGGACTCTGGCTTTGT
*IL-17*	CTCAAAGCTCAGCGTGTCCAAACA	TATCAGGGTCTTCATTGCGGTGGA
*IL-18*	CAGGCCTGACATCTTCTGCAA	TTTGATGTAAGTTAGTGAGAGTGA
*IL-22*	GGTGACGACCAGAACATCCA	GACGTTAGCTTCTCACTTTCCTT

## Data Availability

The data that support the findings of this study are openly available in [repository name “Cells-GDF5-CBFB-Figures” and “Cells-GDF5-CBFB-Figures-Raw data”] at https://drive.google.com/drive/folders/1HzhIw0tjK9N0jeBtw1cG54rShhClxZLh (accessed on 13 February 2023).

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
