# Peer review of "Cbfβ Is a Novel Modulator against Osteoarthritis by Maintaining Articular Cartilage Homeostasis through TGF-β Signaling"

_cells, 2023, doi:10.3390/cells12071064_

Round 1

Reviewer 1 Report

The manuscript entitled “Cbfβ is a novel modulator against osteoarthritis by maintaining articular cartilage homeostasis through TGF-ꞵ signaling” is well written. The authos applied so many techniques to prove their hypothesis. There are some mistakes/deficiensies in the manuscript. I wrote them below for the authors to correct.

 *What do the authors mean by maintaining articular cartilage in abstract lines 20-21

 “.....the role of Cbfβ in maintaining articular cartilage remains obscure.”

 *surgical-induced mice OA model” in abstract lines 25-26 may be changed to “surgically induced OA model (DMM) in mice”

 *The authors sometimes used the term “articular cartilage” and sometimes “joint cartilage”. They should standardize it.

 *The authors wrote “Moreover, the expression of Cbfβ decreased in both mouse and human osteoarthritic cartilage (Fig. 1F-G).” on page 6, lines 222-223. Do the figures show the cartilages of mouse and human in figure 1 f and figure 1g, respectively. The authors should clarify it.

 *Figures 1f, 1g, 2a, 2e, 3a, 3c, 3d are small. The authors should make the figures bigger.

 *The authors should correct “join cartilage” on page 7, line 265.

 *In figure 4c legend, although the authors mention Runx1 qRT-PCR, there is no figure for it.

 *What do the authore mean by HO-control and HO-Tgfb1 in figure 4c?

 *Although the authors wrote 2.4. β-galactosidase (β-gal) staining in Materials and Methods they did not put any result in the results section. 

Author Response

1. *What do the authors mean by maintaining articular cartilage in abstract lines 20-21.  “.....the role of Cbfβ in maintaining articular cartilage remains obscure.” Answer) Thank you for your comment. We correct the sentence as ‘the role of Cbfβ in maintaining articular cartilage integrity remains obscure’

2. * “surgical-induced mice OA model” in abstract lines 25-26 may be changed to “surgically induced OA model (DMM) in mice”

Answer) We changed the sentence to match the reviewer’s comment in the manuscript.

3. *The authors sometimes used the term “articular cartilage” and sometimes “joint cartilage”. They should standardize it.

Answer) We standardized it as ‘articular cartilage’ in the manuscript.

4. *The authors wrote “Moreover, the expression of Cbfβ decreased in both mouse and human osteoarthritic cartilage (Fig. 1F-G).” on page 6, lines 222-223. Do the figures show the cartilages of mouse and human in figure 1 f and figure 1g, respectively? The authors should clarify it.

Answer) Per the reviewer’s suggestion, we indicated mouse and human OA in both Figure 1f, Figure 1g, and legends. We also corrected the sentence as “Moreover, the expression of Cbfβ decreased in mice (Fig. 1f) and human osteoarthritic cartilage (Fig. 1g).”

5. *Figures 1f, 1g, 2a, 2e, 3a, 3c, 3d are small. The authors should make the figures bigger.

Answer) We made the figures larger in the manuscript. Besides, we corrected the mislabeling of Fig. 3c in IHC.

6. *The authors should correct “join cartilage” on page 7, line 265.

Answer) We corrected the “join cartilage” as “Articular cartilage” in the manuscript.

7. *In figure 4c legend, although the authors mention Runx1 qRT-PCR, there is no figure for it.

Answer) Thank you for your comment. We deleted Runx1 In figure 4c legend.

8. *What do the authors mean by HO-control and HO-Tgfb1 in figure 4c? Answer) The HO means Cbfβ△ac/△ac. We revised the image as Cbfβ△ac/△ac.

9. *Although the authors wrote 2.4. β-galactosidase (β-gal) staining in Materials and Methods they did not put any result in the results section.

Answer) We mentioned b-galactosidase in the results section (page 8, line 248). “With Rosa26 reporter (R26R) mice Gdf5-Cre activities were explicitly observed in the articular cartilage by β-galactosidase staining (Supplementary Fig. 2a), with around 75% deletion in articular cartilage by qRT- PCR (Supplementary 2b) and immunofluorescent staining in E16.5 (Supplementary Fig. 2c and 2d) and 20-week-old (Supplementary Figure 1e).”

Reviewer 2 Report

In this article "Cbfβ is a novel modulator against osteoarthritis by maintaining articular cartilage homeostasis through TGF-ꞵ signaling". The authors examined the role of Cbfβ in OA. The results indicated that Cbfβ decreased in aged mouse articular cartilage and human OA cartilage. Articular chondrocyte-specific Cbfb-deficient mice (Cbfb△ac/△ac) exhibited early cartilage degeneration at 20 weeks old and developed OA at 12 months old. Cbfb△ac/△ac mice showed enhanced OA progression under the surgical-induced mice OA model. This work is well designed and provided new concept in OA therapy.

1. The detail information of antibodies used in this study should be provided.

2. The n number of human samples and detail information such as age, gender should be provided

3. Fig 1g, the scale bar should be provided.

4. The limitation should be discussed

5. The further application should be discussed

Author Response

1. The detail information of antibodies used in this study should be provided. Answer) We provided detailed information on antibodies used in this study as follows.

2. The n number of human samples and detail information such as age, and gender should be provided

Answer) We provided detailed information on human OA patients information as follows.

3. Fig 1g, the scale bar should be provided.

Answer) We added the scale bar in figure 1g.

4. The limitation should be discussed

Answer) We added the limitation in the discussion as follows. “This study has a few limitations. How the deletion of Cbfꞵ increased p-Smad3 in the articular cartilage of Cbfb△ac/△ac mice in vivo and primary chondrocytes in vitro is still un-clear. P-Smad3 upregulation may result from the negative feedback loop in the TGF-β signaling pathway or a compensatory mechanism. Unknown factors regulated by or in-volved in the Cbfβ/Runx1 complex in articular chondrocytes need further studies.”

5. The further application should be discussed

Answer) We added the application to the discussion as follows. “Thus, local upregulation of Cbfβ expression using various methods, including small molecules, AAV vectors or other mRNA gene delivery systems in joints, might play a critical role in protecting and delaying cartilage degeneration, which was expected to provide a therapeutic method for OA treatment [47].”

Round 2

Reviewer 2 Report

The authors addressed all my comments. Accept to publish.

Author Response

line 135: MgCl2
write the 2 subscript

Answer) Thank you for your comment. We correct the MgCl2 in the manuscript.

line 166: "24 h treatment with TGF-ꞵ" a table 1 was added summarizing chemicals and antibodies but TGFbeta is missing there. It should be indicated which subtype of TGFbeta was used (throughout the manuscript) or is there any reason not to specify it? I think it is important since the subtypes differ in their effects.

Answer) We added TGF-ꞵ1 information in table 1, and corrected TGF-ꞵ to TGF-ꞵ1 (the subtype of the TGF-ꞵ) in the manuscript.